# Unravelling the Role of Kinases That Underpin Androgen Signalling in Prostate Cancer

**DOI:** 10.3390/cells11060952

**Published:** 2022-03-10

**Authors:** Katie Joanna Miller, Mohammad Asim

**Affiliations:** Department of Clinical and Experimental Medicine, University of Surrey, Guildford GU2 7WG, UK; km00721@surrey.ac.uk

**Keywords:** androgen receptor, kinase, prostate cancer, androgen-regulated genes

## Abstract

The androgen receptor (AR) signalling pathway is the key driver in most prostate cancers (PCa), and is underpinned by several kinases both upstream and downstream of the AR. Many popular therapies for PCa that target the AR directly, however, have been circumvented by AR mutation, such as androgen receptor variants. Some upstream kinases promote AR signalling, including those which phosphorylate the AR and others that are AR-regulated, and androgen regulated kinase that can also form feed-forward activation circuits to promotes AR function. All of these kinases represent potentially druggable targets for PCa. There has generally been a divide in reviews reporting on pathways upstream of the AR and those reporting on AR-regulated genes despite the overlap that constitutes the promotion of AR signalling and PCa progression. In this review, we aim to elucidate which kinases—both upstream and AR-regulated—may be therapeutic targets and require future investigation and ongoing trials in developing kinase inhibitors for PCa.

## 1. Introduction

The biological effects of male sex hormones, androgens, on the target tissues are manifested by the androgen receptor (AR) to achieve a wide range of developmental as well as physiological responses. Thus, androgens play a crucial role in the differentiation and maturation of the male reproductive organs and androgens are the major regulators of cell proliferation and cell death of the epithelial cells within the prostate gland [1,2]. It is, therefore, not surprising that the majority of prostate cancers (PCas) remain reliant on the AR signalling axis throughout their life [3,4,5].

Therefore, to combat PCa, the AR and AR signalling pathways have been considered a key target, and androgen deprivation therapy (ADT) is the current standard of care for advanced PCa. Whilst some ADT reduce androgen synthesis, such as Abiraterone, other drugs, including enzalutamide, target the AR directly. However, in metastatic patients, ADT resistance due to transition to hormone-insensitive castration resistant PCa (CRPC) usually occurs within 18–36 months [6]. There are several resistance mechanisms, including AR amplification, increased AR promiscuity, AR variants (AR-V), and increased reliance on androgen-independent activation of the AR. CRPC can be treated with chemotherapeutic agents such as Docetaxel; however, CRPC is usually fatal within 13–30 months [7]. Notably, ADT-induced activation of the pro-survival poly (ADP-ribose) polymerase (PARP) pathway can also lead to therapy failure and, therefore, combination treatments to simultaneously block AR and PARP signalling hold promise for the treatment of CRPC [8]. Although there have been strives in developing new small molecules which can bind different AR domains reviewed in [9], the identification of other fundamental proteins in the AR signalling cascade to therapeutically target to reduce ADT resistance and treat CRPC is an active and ongoing area of research.

Owing to the fact that AR is a very dynamic phosphoprotein, it is unsurprising that its stability, activity, and cellular localisation are all differentially regulated by phosphorylation by specific signalling kinases [10,11,12]. As key post-translational modifiers in signal transduction, kinases can phosphorylate serine, threonine, and tyrosine amino acid residues within their target proteins. The AR belongs to the family of nuclear hormone receptors and functions as a ligand-activated transcription factor. While unliganded AR is cytosolic, androgen binding triggers the nuclear shuttling of the AR and binding to the androgen response elements (AREs) motifs that are often present in the promoters and enhancers of the AR target genes, including many kinases which mediate changes in cell proliferation and metabolism [13]. In several cases, downstream kinases also directly interact with the AR, resulting in feedback regulation of AR activity [14]. The following manuscript will review current knowledge on kinases around AR signalling and explore their role as potential therapeutic targets for the treatment of PCa.

### 1.1. AR as a Master Transcription Factor

The AR is overexpressed in PCa as a function of disease progression, with ADT providing a selection pressure to overcome androgen dependence and develop into advanced disease such as castration-resistant prostate cancer (CRPC) [3]. Thus, the AR is a major therapeutic target throughout PCa progression. The unstimulated AR resides in the cytosol where several chaperones interact with the AR to oppose its degradation by cytosolic proteases. However, once activated, the AR translocates from the cytoplasm to the nucleus, where it binds to a cis-regulatory consensus sequence with the DNA known as Androgen Response Elements (AREs) [15]. Not all androgen-regulated genes have canonical AREs in the proximal promoter; however, AREs may be found in the distal enhancer regions [16,17], or AR recruitment may otherwise be supported by transcription ancillary factors such as Sp1 and ETS [18,19,20]. While the major function of the AR is to act as a transcriptional activator (i.e., a transactivator), a chromatin-bound AR can also act as a repressor of the target gene expression. When acting as a transactivator, AR recruitment creates a platform for RNA Pol II machinery to bind, which increases the rate of transcription. TMPRSS2, PSA, and PSCA are all examples of genes whose expression is upregulated by AR in this manner [21,22,23]. Transrepression by the AR is less well understood. An example of AR-mediated suppression of the expression of Maspin gene transcript level. The AR and Maspin form a negative feedback loop where each bind to consensus sequences in the other’s promoter to reduce RNA Pol II binding and decrease transcription [24,25].

In addition to its primary role as a transcription factor, a few non-genomic (transcription-independent) functions of the AR have been revealed in the recent years, including the binding of AR with a scaffold protein filamin A in prostate fibroblasts to promote oncogenesis [26], and steroid-induced association of AR with oestrogen receptor beta and Src to promote cancer cell proliferation [27].

### 1.2. Functional Domains of the AR

The functionality of the AR is primarily split into three well-defined functionally critical domains; the amino terminus transactivation domain (NTD), the DNA binding domain (DBD), and the ligand-binding domain (LBD), with a hinge region connecting the LBD and DBD (Figure 1). All these domains can act as targets of small molecules designed to inhibit AR function in prostate cancer reviewed in [9]. The transactivation potential of the AR primarily involves the NTD, which contains activation function 1 (AF1) that harbours two transactivation units (TAU), TAU1 and TAU5 [28]. The AF1 is constitutively active on its own; however, its interaction with the LBD prevents the action of the receptor without an androgenic ligand [29]. These interactions are lost in clinically relevant AR variants (AR-Vs) that lack the LBD potentially making AR-Vs to become constitutively active. Androgen binding to the LBD results in a conformational change, which exposes the activation function 2 (AF2) within the LBD allowing it to bind to transcriptional cofactors to modulate gene transcription [30]. The conformational change also causes the chaperones to dissociate from the AR and the nuclear shuttling of the AR [30]. In the nucleus, the AR interacts with DNA through its DBD, exploiting the zinc finger motifs in the P-box and D-box, which recognise AREs [31]. The DBD is highly conserved between nuclear hormone receptors, and therefore, may not represent a promising therapeutic target due to potential off-target effects [32].

Another layer of intricacy to AR-dependent transcription is the involvement of the co-regulatory proteins, such as coactivators that can bind to the AR to enhance the rate of AR-mediated transcription, and transcriptional corepressors—which usually bind to the NTD—which can dampen AR transactivation function causing a reduction in AR target gene expression. Two main conformations of the AR have been proposed by which AR can regulate, which co-regulatory molecules bind: the ligand-bound conformation of the LBD and post-translational modifications, such as phosphorylation [11,34]. The binding of an agonist such as androgens can modulate AR conformation that allows preferential recruitment of transcriptional coactivators such as CBP/p300 and SRC [35]. In contrast, when an androgen antagonist such as cyproterone acetate (CPA) binds to the LBD of AR, the conformational change preferentially allows corepressors, such as Alien and Silencing-Mediator for Retinoid/Thyroid hormone (SMRT) to be preferentially recruited [36].

Phosphorylation of the AR by kinases is known to occur in at least 15 amino acid residues within the AR protein (Figure 1) and can potentially modulate which coregulators bind, as well as AR stability and location reviewed in [37,38]. In AR-Vs, such as AR-V7, which lack a functional LBD, ligand-dependent activation is lost whilst phosphorylation of the remains a key form of AR regulation [39]. Therefore, kinases that enhance AR-mediated transcription can represent surrogate therapeutic targets to stop constitutively active AR-Vs function. Given that the NTD is intrinsically disordered, anecdotal compound screening techniques will have to be used to find potential inhibitors rather than designed antagonists based on the structure [9], therefore targeting of kinases that modulate the NTD may prove more successful in tumours that are driven by such AR-Vs.

## 2. Upstream Kinases of the AR

Components of several key signal transduction pathways have been shown to interact with androgen signalling, including direct phosphorylation of the AR, through their kinase components. These include the IL-6 and EGF extracellular signal-mediated pathways and the PI3K/Akt and MAPK transduction networks, which are often found to be implicated in hormone-independent aggressive PCa such as CRPC [40]. Likewise, cell cycle-dependent kinases can also phosphorylate and regulate AR activity to allow cell proliferation. Within each of these pathways, there are key kinases that can act as potential therapeutic targets for cancer inhibition.

### 2.1. Interleukin 6 Pathway

Interleukin 6 (IL-6) is a pleiotropic cytokine that is involved in the regulation of the growth of the majority of malignant tumours [41], including PCa [42]. Exposure of PCa LNCaP cell line to Bicalutamide—an AR LBD antagonist—resulted in a 70% decrease in the IL-6 induced expression of AR target gene PSA, indicating the role of IL-6 signalling in promoting AR activity [43]. The role of IL-6 in regulating AR activity appears to be pleiotropic since several signal transduction cascades can be activated by the binding of IL-6 to its receptor, including the JAK/STAT, MAPK, PI3K/Akt, and PKA pathways, providing multiple possible mechanisms for the modulation of AR signalling. Using kinase inhibitors to disrupt these signalling pathways, IL-6 mediated activation of the NTD of AR was shown to occur primarily via JAK/STAT and MAPK pathways [43]. This IL-6 mediated activation of AR may occur via STAT3, which interacts with amino acids 234–558 of the AR NTD [43] and requires phosphorylation of STAT3 at Tyr705 and Ser727 for optimal activation [44]. JAK can phosphorylate STAT3 at Tyr705 [45] and kinases, such as p38MAPK and ERK in the MAPK pathway, can also phosphorylate STAT3 at Ser727, thus promoting STAT3-mediated AR activation [46,47]. Due to its role in IL-6 mediated signalling, the role of JAK has been explored as a potential therapeutic target to block PCa progression; however, this has proved a challenge. Ruxolitinib is a JAK1/2 inhibitor that reached Phase II trials for metastatic PCa, but the trial was terminated due to low efficiency (Trial identifier: NCT 00638378); however, this should be further investigated in the context of prolonged ADT with antiandrogens [48], where IL-6 signalling is more prevalent [49]. The Jak inhibitor AZD1480 initially also showed promise, suppressing metastasis in pre-clinical models [50]; however, three Phase I trials have been terminated due to unfavourable neurotoxicity profiles [48]. Another stalling factor in JAK inhibitor development may be a lack of clinically representative cell-based assays [43]. Src kinase is a non-receptor tyrosine kinase at the confluence of several signal transduction pathways, such as EGFR, IL6, MAPK, and PI3K [51]. Like JAK, Src can also phosphorylate STAT3 at Tyr705—albeit much weaker [52]—and promotes non-genomic AR signalling; however, beyond this, it also directly interacts with AR between residues 371 and 381 in the AF1 in the NTD domain [53]. Src is not to be confused with steroid hormone coactivators (SRC), which also interact with the AR and are required for ligand-independent activation by IL-6 [54]. Through the feed-forward regulation, the AR-Src interaction activates the Src [53], which results in Src-mediated phosphorylation of the AR at Tyr534. This increases AR stability by blocking E3 ligase CHIP and the subsequent proteasomal degradation [55], as well as causing recruitment p85α and the resultant activation of MAPK and PI3K/Akt pathways [37]. Src can also indirectly activate AR-mediated transcription by repressing the AR interaction of AR with AR corepressor LCoR [56]. Dasatinib inhibits Src and has shown promising results in Phase II trials of decreased metastasis; however, another Phase II trial did not improve on abiraterone, and in Phase III trials did not improve on docetaxel [51,57,58]. It has been suggested that identifying which subpopulation of patients have higher Src-driven PCa would be beneficial for future Dasatinib trials [58]. One such subpopulation may be AR-V7-driven CRPC [58,59].

### 2.2. Epidermal Growth Factor Pathway

There is also crosstalk between several growth factors signalling pathways and the AR signalling pathway, with the epidermal growth factor (EGF) receptor family amongst the most described in the literature. EGF pathways play a pivotal role in regulating AR signalling, e.g., ERBB1 (EGFR) expression is enhanced as a function of tumour severity, whilst the expression of ERBB2 (HER2) increases during and following androgen ablation, thus ensuring sustained survival of PCa cells [60,61].

There is functional redundancy in the activation of downstream signalling by IL-6 and EGF signal transduction pathways. Like IL-6, EGF signalling can also promote phosphorylation of STAT3 at Tyr705, although this occurs via the EGFR and Src [62,63]. EGFR signalling also cross talks with the AR synergistically during ligand-dependent AR activation via activation through the use of the ERK1/2 and p38 MAPK pathways [64], possibly via Ser515 phosphorylation which transactivates the AR and is involved in AR recycling [65]. The EGFR can also activate the protein kinase C (PKC) pathway, which phosphorylates Ser578 of the AR, which causes greater transactivation than Ser515 phosphorylation [65]. Notably, Ser578 is within the P box of the DBD, and its phosphorylation promotes AR-DNA interaction and specificity [65]. Intriguingly, phosphorylation of AR Ser515 and Ser578 are coregulated, with phosphorylation of P-Ser578 correlated with lower P-Ser515; this coregulation regulates AR nuclear-cytoplasmic shuttling through interactions with the Ku-70/80 [65].

EGF stimulation of the EGFR and HER2 receptors results in phosphorylation and activation of activated Cdc42-associated tyrosine kinase 1 (ACK1) [66,67], which can phosphorylate the AR and enhance transcription of the AR gene itself. ACK1 kinase phosphorylates the AR at Tyr267 and Tyr363, located in the AF1, to promote AR-mediated transactivation of AR-regulated genes, with Tyr267 being the functionally important site [67]. P-Tyr267 is required for the AR to bind to the Ataxia telangiectasia mutated (ATM) enhancer, and the consequential increase in ATM expression maintains genetic integrity from double-stranded DNA breaks. This correlates with PCa progression and affords greater resistance of CRPC to radiotherapy [68]. ACK1 can also phosphorylate Akt at Tyr176, which promotes Akt translocation to the plasma membrane and subsequent activation of Akt and the PI3K pathway [69]. ACK1 activation also results in an increase in AR expression via phosphorylation of histone 4 (H4) at Tyr88. P-Tyr88-H4 has been located at two sites upstream enhancer sites of the AR, creating an area of permissive chromatin to promote AR mRNA transcription in CRPC [70]. Therefore, ACK1 activation may be involved in AR transactivation and in the increase in AR expression in PCa to overcome androgen deprivation therapy (ADT) and the progression to lethal CRPC.

In normal prostate epithelia, the EGF pathway mediated phosphorylation of ACK1 results in the degradation of ACK1 and EGFR in a negative feedback mechanism, mediated via the interaction of E3 ligases with the ubiquitin association (UBA) domain of the ACK1. This appears to be overcome in CRPC, as both ACK1 and EGFR protein levels are increased, presumably due to changes in phosphorylation and protein interactions in the ACK1 UBA domain [71]. The reduced ACK1 degradation may allow for the previously mentioned oncogenic effects of ACK1 activation to be exacerbated.

Due to the role of ACK1 in AR transactivation and PCa genomic integrity, several inhibitors have been designed that show potential to overcome CRPC [66]. Notably, Dasatinib, previously mentioned for its role as an Src inhibitor, also inhibits ACK1 [72]. AIM-100 is more selective for ACK1 and inhibits LNCaP and LAPC-4 cell proliferation; however, poor pharmacokinetic properties have halted entry into clinical trials [73].

### 2.3. PI3K/Akt Signalling

The PI3K/Akt signalling network is involved in complex crosstalk between several cell signalling cascades, some of which have the potential to be oncogenic and have already been mentioned earlier. Two key points in the signalling network are mTOR1 and mTOR2 both of which are functionally distinct complexes downstream of PI3K/Akt, which regulate many cellular processes that promote tumour progression and drug resistance [74]. The intertwining branches in the PI3K/Akt signalling network contains many redundancies and feedback regulatory loops, which results in limited efficacy in clinical targeting. However, more promising results with the use of their inhibitors have been achieved in recent years, reviewed in [74].

PI3K/Akt/mTOR and AR signalling pathways have a complex relationship containing several redundancies, and downregulation of one can promote the other; this can occur during AR inhibitor therapy, where downregulation of AR-regulated genes can result in increased AKT-mTOR signalling and lead to therapeutic resistance [75]. Conditional activation of Akt in LNCaP cells promoted proliferation in absence of androgens, which was confirmed in transgenic mice, suggesting a role of the Akt pathway in the transition from androgen dependence to androgen independence [76]. These data suggest that Akt is an attractive therapeutic target for growth inhibition, in particular when PTEN—a tumour suppressor phosphatase that negatively regulates the PI3K/Akt pathway—is inactive or deleted, which occurs in up to 60% of CRPC [77]. Inhibition of both Akt and AR signalling pathways has been considered therapeutically in PTEN-deficient PCa in the hope of inducing synthetic lethality. Recently, the Akt inhibitor Ipatasertib was combined with the Abiraterone—an inhibitor of androgen biosynthesis—in Phase III trials. Encouragingly, this study showed an improved radiographical progression-free survival [78], indicating that PI3K/Akt blockade alongside ADT or AR inhibition is a viable option in the treatment of advanced PCa.

Growth factor and cytokine pathways mentioned previously activate kinases, such as Src and ACK1, which positively regulate the activity of both the AR and Akt, and therefore, activation of these membrane-associated receptors may be key in the development of therapeutic resistance and progression to CRPC. However, under low androgen conditions, the AR and Akt can also directly interact, with Akt phosphorylating the AR at Ser213 and Ser791 [79]. In vitro, the outcome of this interaction is varied and may depend on different cellular contexts, including the passage number of the PCa cells, with Akt enhancing AR activity in high passage cells but reducing AR activity in low passage cells [76].

### 2.4. Mitogen-Activated Protein Kinase Pathway

The activation of the MAPK signal transduction pathway is yet another mechanism that growth factors employ to activate PCa growth. The MAPK pathway is split into four parts, the extracellular signal-regulated ERK1/2 pathway, the EGF- and oxidative stress-associated ERK5 pathway, and the stress-activated JNK and p38 MAPK pathways. Androgen-mediated AR signalling can induce the ERK1/2 and p38 MAPK pathways [64], which can then mediate proliferation via phosphorylation of several transcription factors such as Myc [80,81]. Activation of the ERK1/2 pathway can also occur via IL-6 and growth factor signalling [64,82], and phosphorylation of ERK correlated with the severity of PCa [81,82,83]. The primary ERK1/2 kinases are MEK1/2, and MEK1/2 inhibition is being investigated as a potential avenue to dampen AR signalling [84,85]. Trametinib, a MEK1/2 inhibitor that is approved therapeutic for melanoma, is currently undergoing Phase II trials for CRPC (Trial identifier: NCT02881242).

Expression and activation of p38 MAPK and some of its associated upstream kinases occurs in PCa, making this pathway another possible therapeutic target [86]. Interestingly, androgen-driven activation of p38 MAPK results in the phosphorylation of AR-interacting heat shock protein 27 (hsp27), which subsequently replaces hsp90 as an AR chaperone for translocation into the nucleus and aids AR transactivation [87]. IL-6 can also activate p38 MAPK signalling independent of androgen, which by this mechanism would increase AR signalling [87]. Inhibition of p38 MAPK signalling decreased proliferation in AR-positive cell lines in normoxia, and growth was further compromised by hypoxia, which is another trigger for p38 MAPK activation [88].

### 2.5. Cell Cycle-Dependent Modulation of AR Activity

AR signalling results in the proliferation of PCa epithelial cells via regulating the transition between the G1 and S phases [89,90]. At the beginning of the G1 phase, AR activation induces mTOR-dependent translation of D1 cyclin, resulting in cyclin D1 accumulation and subsequent interaction with the Cyclin-dependent kinase (CDK)4/6 complex [91]. Activated D-cyclin/CDK4 or CDK6 complex phosphorylates and inactivates retinoblastoma tumour suppressor (Rb) in G1, causing it to release E2F transcription factors [92]. Activated E2F1 transcription factor not only enhances AR expression but also causes increased expression of cyclin A and activation of CDK2 for transition into S phase [93].

Many ADT rely on Rb [58,92], and thus, if Rb activity is lost due to deletion, mutation, or PTEN inactivity [26,27], the antiandrogen therapy can be compromised [92]. Indeed, Rb deletion is higher ADT resistant CRPC than in primary PCa [93]. Beyond the cell cycle, Rb loss changes AR transrepression activity and can alter the AR cistrome [94].

AR activity is also modulated through phosphorylation by other CDK throughout the cell cycle. Firstly, Ser81 of AR can be phosphorylated by CDK1, CDK2, CDK5, and CDK9 [11,95]. Phosphorylation of AR at Ser81 is associated with androgen-mediated activation of the AR [96]. Depending on the context of phosphorylation, P-Ser81 of AR results in resistance to AR degradation, increased nuclear localisation and chromatin binding, as well as interaction with cofactor CBP/p300 and expression of different subsets of AR-target genes [97,98,99]. CDK1 is active in the G2/M phase, and phosphorylation of Ser81 of AR in cytoplasm causes AR translocation to the nucleus whilst preventing degradation and may facilitate the transcription of AR target genes following mitosis [100]. As opposed to CDK1, the role of CDK9 is not cell cycle-dependent; however, once AR is recruited on its target genes, CDK9 can phosphorylate its Ser81 to couple the P-TEFb transcription elongation complex that activates RNA Pol II to synergise AR-dependent transcription [97,99,101]. CDK5, as well as promoting stability and transactivation of AR by phosphorylating Ser81, also promotes degradation of p21^CIP1^, an inhibitor of CDK2/3/4/6 [96,102]. CDK2 can also phosphorylate Ser81 of AR [103], and CDK6—beyond its role in Rb phosphorylation—can interact with the AR to promote transactivation [104]. The other main CDK in the cell cycle is CDK11, specifically the CDK11p58 isoform, which codes for a transiently active kinase in mitosis associated with spindle formation and cell cycle arrest [105,106]. The CDK11p58/cyclinD3 complex causes the phosphorylation of the AR on Ser308, in TAU1 within the NTD, which results in the exclusion of the AR from condensed chromatin and the repression of AR activity [107]. However, a therapy to promote CDK11p58 activation to reduce AR activity may not be a good strategy as CDK11—the isoform was not reported—has been reported to promote AR expression and activation in osteosarcoma [108]. This would be a concern, as the bone is the most common PCa metastasis site [109], and thus, increasing the activity of CDK11 may be harmful in metastatic CRPC. The majority of interest in CDK inhibitors for PCa currently lies with CDK4/6, rather than AR Ser81 kinases. Three CDK4/6 inhibitors—abemaciclib, palbociclib, and ribociclib—have been found safe and effective in hormone receptor-positive HER2-negative breast cancer and are now undergoing clinical trials for various stages of PCa reviewed in [110,111]. However, abemaciclib is also a CDK9 inhibitor [112], and several selective small-molecule CDK9 inhibitors have been designed. KB-0742 has been designed as ‘ultra-selective’ for CDK9 and showed promise by inhibiting AR-V7 dependent transactivation in CRPC xenografts [113], and has entered Phase I clinical trials (Trial identifier: NCT04718675).

## 3. Downstream Kinases of the AR

Normally, the AR acts as a transactivator for target genes leading to their increased expression. The AR can also repress gene expression by acting as a transrepressor. These target genes code for proteins with a variety of roles, including mediation of cell proliferation and metabolism. Some AR target genes code for downstream kinases which can reinforce AR signalling by interaction with the AR to change its stability, location, and transactivation by phosphorylation of key serine, threonine, and tyrosine residues. On the other hand, androgen-repressed kinases are also therapeutically important, as these may function as backup pathways that are activated in response to AR-targeting therapies and can mediate therapeutic resistance. We identified 49 androgen-regulated kinase genes in LNCaP cells, of which 25 were upregulated and 24 were downregulated [14], which are discussed further below (see Figure 2). Massie et al. partially defined AR target genes as having ARE within 25 kb of a gene, which, although allowing enough genes for pathway enrichment analysis, would have missed some genes regulated by more distal ARE [13], which may encode proteins that could present as potential therapeutic targets. Indeed, three AR-upregulated and sixteen AR-downregulated kinases identified by Asim et al. did not have ARE within 25 kb [14]. Protein interaction networks and clustering analysis of the 49 kinases highlighted clusters of functional enrichment, including growth factor signalling, MAPK signalling, and glycolysis [14]. To identify any other biological processes that increased or decreased after androgen signalling, Gene Ontology (GO) analysis was performed on the forty-nine kinases. GO analysis of upregulated kinases also indicated enrichment for regulation of the MAPK pathway, regulation of stress-activated MAPK pathway, and positive regulation of PI3K signalling. GO analysis of the downstream kinases implicated nuclear speck organisation, sodium and potassium iron transport across the membrane, and regulation of the ERK1/2 cascade.

### 3.1. Cell Metabolism

Cell metabolism is a broad term for biochemical reactions that are required to sustain a cell and consists of several overlapping groups of enzymes. A number of these enzymes are kinases, several of which are androgen regulated (Figure 2). Whilst some of the kinases are more specific to certain cellular processes, such as glycolysis, others are key regulators of metabolism.

#### 3.1.1. Glycolysis

AR signalling has been shown to upregulate aerobic glycolysis and anabolic synthesis through upregulation of several enzymes including key rate-limiting kinases in the glycolytic pathway [13]. Hexokinase 2 (HK2) and ADP-Dependent Glucokinase (ADPGK) catalyse the first obligatory step of glucose metabolism in the cytosol and endoplasmic reticulum, respectively. Whilst HK2 utilises ATP, ADPGK utilises ADP and, thus, promotes glycolysis under nutrient-poor and anoxic conditions, such as in cancerous cells, and may support the Warburg effect [114]. AR signalling also increases 6-Phosphofructo-2-Kinase/Fructose-2,6-Biphosphatase 2 (PFKFB2) expression, although PFKFB2 activity can also be increased through phosphorylation of Ser446 and Ser483 by the PI3K/Akt signalling pathway [115]. Therefore, potentially in high passage cancer cells PI3K/Akt and AR signalling may synergistically [76] to increase glucose metabolism. By upregulating HK, ADPGK, and PFKFB2, AR signalling results in increased energy production via glycolysis and carbon is produced for macromolecule synthesis [13]. Although metabolic pathways are potential targets, these three glycolytic kinases are relatively ubiquitously expressed and may, therefore, be poor therapeutic targets for kinase inhibitors to achieve specificity. However, the increase in glucose uptake has been utilised clinically in fluorodeoxyglucose PET scans, used to identify if metastasis has occurred [116].

#### 3.1.2. CAMMK2 Signalling

Calcium/Calmodulin Dependent Protein Kinase Kinase 2 (CAMKK2) is a metabolic regulator, which is targeted by the AR in both androgen-dependent and androgen-independent PCa cell lines [13]. In PCa, CAMKK2 primarily acts through phosphorylation of AMP-activated protein kinase (AMPKα) [13], which in turn phosphorylates many kinases. Amongst AMPKα phosphorylated kinases are PFKFB2/3, which leads to a further increase in Glycolysis [117]. AMPKα also phosphorylates p21 Activated Kinase 2 (PAK2) at Ser20 to activate it [118]. PAK2 is repressed by AR signalling, but its level increases during ADT and has been identified as a ‘hub’ protein for metastasis development [119,120], indicating an oncogenic role for AMPKα in PCa. Knockdown of CAMKK2 reduces transitioning of cells from G1 to S phase of the cell cycle, as CAMKK2 is required for AR-mediated proliferation [121]. Expression of CAMKK2 increases with Gleason Score [121]. The CAMKK2 inhibitor STO-609 has been used in most CAMKK2 studies; however, STO-609 can also repress many other kinases [122]. Even so, treatment of AR-dependent cell lines with STO-609 has been shown to decrease their proliferation, migration, and invasion [13,121,123], indicating the viability of CAMKK2 as a potential target in PCa. The repression of the prostate tumour xenografts was also more pronounced in chemically castrated mice when they were administered with STO-609, despite its poor solubility in vivo [124]. However, this potentially could be the accumulated effect of several inhibited kinases [122]. The process to develop small-molecule inhibitors is ongoing, with the hinge region of CAMKK2 as the main target for achieving CAMKK2 inhibition [122,125,126].

Calcium/calmodulin-dependent protein kinase type 1 (CAMK1) is a transcriptional target of AR [13,14]. However, CAMKK2 inhibition did not change the phosphorylation status of CAMK1, and inhibition of CAMK1 had marginal effects on cell proliferation and aerobic glucose metabolism [13]. Therefore, whilst CAMKK2 has been investigated as a therapeutic target in PCa, CAMK1 has not and may be interrogated for its role as a therapeutic target in PCa.

#### 3.1.3. Choline Kinase Alpha

Choline kinase alpha (CHKA) is an androgen-upregulated kinase that has roles in lipid metabolism and positive regulation of AR activity [14,127]. In lipid metabolism, CHKA catalyses the first step in the Kennedy pathway for the synthesis of phosphatidylcholine, a key component of cell membranes [127]. Furthermore, phosphorylation of CHKA at Ser279 by AMPKα in low glucose conditions increases lipolysis, which provides substrates for beta-oxidation, a key process for tumour growth [128]. As AMPKα is activated by CAMKK2, CHKA may play a role in CAMKK2-mediated alteration of metabolism. In addition to PCa, CHKA is oncogenic, and it is known to be upregulated in many cancer types and may, thus, represent an attractive therapeutic target [129,130,131].

However, CHKA is of particular therapeutic interest in PCa due to its interactions with AR both directly and indirectly. CHKA acts as an AR chaperone by binding to the LBD and maintaining AR stability and results in a feed-forward signalling loop that maintains AR activity [14]. However, although CHKA knockdown reduced full-length AR transcriptional activity, ARv567—a variant of AR—was not inhibited via CHKA knockdown [14]. CHKA may indirectly impact the AR by aiding in the activation of EGF-mediated signalling via interaction with the EGFR, which is Src-dependent [132]. Therefore, CHKA may be important for the progression of PCa to CRPC. Furthermore, the interaction between EGFR and CHKA is required for EGF-mediated DNA synthesis and increases proliferation, which may a mechanism by which CHKA promotes oncogenesis in many cancers [132]. CHKA inhibition may potentially reduce lipolysis, AR stability, and EGFR signalling, and therefore, presents as a key therapeutic target for investigation in PCa. Small-molecule inhibitors and small interfering RNA (siRNA) have been designed for CHKA inhibition reviewed in [131], one of which—TCD-717—has finished Phase I clinical trials for solid tumours (Trial identifier: NCT01215864), although no results have been posted. A suggested mechanism of TCD-717 action is inhibition of CHKA Tyr333 phosphorylation [133], an amino acid residue that is phosphorylated by Src to allow CHKA interaction with EGFR [132]. As yet, it is unknown if TCD-717 will affect the chaperone function of CHKA.

#### 3.1.4. Coenzyme A Synthase

Coenzyme A synthase (COASY) was identified as upregulated by AR signalling [14]. COASY is a bifunctional enzyme with a synthase and a kinase domain, which catalyse the last two steps in the biosynthesis of CoA—a key metabolite in many biological processes reviewed in [133,134]—from pantothenic acid [135]. The lack of ARE within 25 kb may have excluded COASY from many studies of AR-regulated genes, and literature on the role of COASY within PCa is currently very limited. However, there is indirect evidence of COASY being an AR target gene, and that through the analysis of metabolites, CoA synthesis was identified as a key metabolic pathway changed in androgen ablation therapy [136]. Beyond synthesising CoA, COASY also has a role in the regulation of mitotic proteins, including Aurora A through association with CBP, and DNA repair [137]. COASY knockdown reduced activity of the PI3K pathway and DNA repair, which sensitised colorectal cancer to radiotherapy in vivo and in vitro [138], and thus, may also be a therapeutic target alongside radiotherapy and chemotherapy in CRPC. Although COASY inhibition did not alter proliferation in triple-negative breast cancer and colorectal cancer cell lines [138,139], the outcome in PCa may be different for several reasons. Firstly, COASY activity is strongly upregulated by phosphatidylcholine [135], in which CHKA—an AR-regulated gene [14]—is key for the synthesis of [127]. Thus, COASY may potentially be more active and depended upon in PCa. COASY inhibition may also indirectly impact the phosphorylation and stability of the AR. COASY inhibits Aurora A—a kinase that promotes AR degradation [140]—firstly in mitosis through reduction of CBP acetylation [137], and secondly through interaction with CoA during oxidative stress [141]. Furthermore, the COASY protein also interacts with PI3K-P85α, a regulatory subunit of PI3K, which leads to increased Akt phosphorylation [138] and in turn phosphorylation of the AR [79]. The impact of COASY inhibition of AR-driven proliferation and PCa is currently unknown and requires investigation in vivo and in vitro.

### 3.2. Growth Factor Signalling

As mentioned previously, growth factor signalling can crosstalk with AR signalling and can contribute to progression to androgen-independent and even AR-independent PCa. AR signalling increases EGFR and IGF1R transcription [142,143], and although in normal prostate epithelium EGFR is degraded by prolonged AR signalling [64], this can be overcome by various mechanisms including increased autocrine and paracrine EGF expression [144,145] and potentially changes in ACK1 UBD interaction [71]. Despite its increased expression with severe disease, EGFR is not a key signalling pathway that confers androgen independence [146]. On the other hand, HER2 is post-translationally repressed by the AR [143], and ADT removes this repression leading to its increased in situ expression in advancing disease and tumour recurrence [147]. Therefore, HER2 inhibition is potentially a better therapeutic target than EGFR for CRPC.

### 3.3. MAPK Signalling

One of the functional clusters identified in Asim et al. was MAPK signalling [14]. MAPK signalling is split into four branches, and each branch can be modulated by androgen signalling (Figure 2). Therefore, the level and activity of various MAPK proteins can be modulated not only by the AR inhibition but also as a function of tumour progression.

#### 3.3.1. ERK1/2 MAPK

AR signalling potentiates ERK1/2 MAPK activity by inducing and enhancing the ERK1/2 MAPK pathway, partially through genomic transactivation and repression. Firstly, EGFR and IGF1R—which transduce signals via ERK1/2 signalling—are upregulated [142,143]. There are also several AR-repressed kinases we identified [14] which can negatively regulate the ERK1/2 MAPK cascade, such as Connector Enhancer Of Kinase Suppressor Of Ras 3 (CNKSR3) [148,149], WNK Lysine-Deficient Protein Kinase 2 (WNK2) [150], and Calcium/Calmodulin-Dependent Protein Kinase II Inhibitor 1 [151,152]. CNKSR3 codes for MAGI1, which is a Membrane-Associated Guanylate Kinase, and its depletion by RNAi can increase MEK1/2 phosphorylation and, thus, its activity [149,153]. WNK2 depletion also increases MEK1 activity and sensitivity to lower concentrations of EGF [154]. Repression of CAMK2N1 reduces inhibition of CAMK2 signalling, which could lead to increased MEK1 phosphorylation [155]. Therefore, repression of these three genes—which all encode proteins that reduce phosphorylation of MEK1/2—allows increased activation of the ERK1/2 MAPK pathway.

#### 3.3.2. ERK5 MAPK

The ERK5 MAPK pathway is also activated by growth factors—as well by mitogenic signals and oxidative stress [156]—however, its primary activator Mitogen-Activated Protein Kinase Kinase 5 (MAP2K5) and a further upstream kinase WNK Lysine-Deficient Protein Kinase 1 (WNK1) [157] are downregulated by AR signalling [13]. During ADT, the ERK5 pathway is no longer repressed by AR signalling and has been implicated as a key mechanism underlying AR inhibitor resistance [158], as well as PCa proliferation and metastasis [159,160]. ERK5 inhibitors—such as XMD8-92—have been investigated in other cancers alongside other MAPK inhibitors to combat resistance mechanisms [158,159].

#### 3.3.3. P38 MAPK

The AR-mediated genomic regulation of the pro-survival p38 MAPK appears to be varied. MAP2K3 and MAP2K6 are the primary activators of p38 and are downregulated. Conversely, the AR-mediated increase in Homeodomain Interacting Protein Kinase 2 (HIPK2) [13,14] and Serine/Threonine Kinase 39 (STK39) [14,161] would promote the p38 MAPK pathway. HIPK2 enhances the degradation of WIP1 [162], an inhibitor of p38 MAPK under oxidative stress conditions [163], and the STK39 encoded protein SPAK can activate p38 MAPK [161]. However, the increased expression of HIPK2 and STK39 may not be reflected in the activation level of the encoded protein. HIPK2 has a short half-life [164] and an autoinhibitory domain, but is stabilised and cleaved for activation during DNA damage [165]. HIPK2 can impact AR signalling, but only in low hormone conditions [166]. SPAK has lower phosphorylation of the Ser385 activation residue in hormone naive than metastatic PCa [167]. Whilst HIPK2 and STK39 can activate the p38 MAPK pathway, the reduced expression of MAP2K3 and MAP2K6—as primary activators of the p38 MAPK pathway—are likely to have a much greater impact on p38 MAPK signalling. Therefore, the overall result is likely to be androgen-mediated downregulation of the p38 MAPK pathway.

During ADT, the genomic repression of the p38 MAPK pathway would be ablated and reversed. One of the results would be increased hsp27, which would be predicted to increase stability and nuclear translocation of the AR [87,88], as well as enhance survival via the p38 MAPK pathway [88]. Activation of the p38 MAPK pathway has also been linked to epithelial-mesenchymal transition (EMT) and metastasis [88,168]. By identifying MAP2K3 and MAP2K6 repression by the AR, continuing to repress these alongside ADT could reduce androgen independence caused by the increase in the p38 MAPK pathway. However, only a few MEK3/6 inhibitors have thus far been designed [169,170].

#### 3.3.4. JNK and MAP2K4

MAP2K4 is another key MAPK gene that we and others have found to be transcriptionally regulated by the AR [13,14,171]. MAP2K4 codes for MEK4, which can activate the JNK and p38 MAPK pathways. Both of these pathways result in the phosphorylation of AR on Ser650 in its hinge region, leading to AR translocation from the nucleus into the cytoplasm, therefore decreasing AR transactivation [172]. MEK4 is primarily involved with JNK-mediated Ser650 phosphorylation [173]. The AR-mediated MEK4 increase, thus, forms a negative feedback mechanism on AR signalling. Although this implies that the JNK pathway should be repressed during ADT, this has not been reported. Indeed, overall JNK has been reported as repressed by androgen signalling [173], and therefore, JNK levels would be predicted to increase following treatment with enzalutamide [174], genipin [174], and guggulsterone [175], and can, thus, cause apoptosis. Conversely, JNK activation is also implicated in enzalutamide-mediated metastasis [176]. This builds a contrary picture of the role of the JNK MAPK pathway in prostate cancer progression. It should be noted that whilst MEK4 is required for optimal JNK activation, MEK7—which phosphorylates a different JNK residue—is essential JNK activation [177]. To our knowledge, MEK7 is not an AR-responsive gene. Thus, whilst a decrease in MEK4 due to ADT may reduce, JNK activation is primarily activated by a non-AR target gene.

Although targeting MEK4 may have limited effects in modulating the JNK MAPK pathway, MEK4 is still targeted for the activation of the p38 MAPK pathway, which increases AR stability by hsp27 expression and metastasis via matrix metalloproteinase-2 (MMP-2) expression. However, chronic overexpression of MAP2K4 in the LNCaP cell line indicated MEK4 can also bypass p38 MAPK and increase the stability of hsp27 and MMP-2 expression [178]. Phase II trials of the MEK4 inhibitor Genistein decreased prostate cancer metastasis in the short term; however, compensatory mechanisms may reduce the efficiency of long-term MEK4 inhibition [179].

### 3.4. CDK8 and the Mediator Complex

CDK8 has been identified as an AR-repressed gene [180,181]. Whilst CDK8 expression remains unchanged between benign prostate tissue and primary PCa, it is elevated in ADT-treated metastatic and CRPC [182]. CDK8 is a component of the mediator complexes [183] and opposes the interaction of the mediator complex with the preinitiation complex at a gene promoter and RNA Pol II, inhibiting transcription [184]. However, during stressful conditions, potentially including ADT and CRPC which have higher oxidative stress than those without PCa [185,186], CDK8 becomes active and dissociates from the mediator complex to allow increased transcription [187,188].

In the VCaP cell line—which is androgen-dependent and represents primary PCa [16]—the inhibition of CDK8 increased transition from G1 to S phase [189], possibly due to the increase in transcription-promoting mediator complexes without CDK8. Thus, androgen-mediated repression of CDK8 in primary PCa may also promote G1 to S phase transition. Although CDK8 expression is increased in ADT and CRPC [182], this may not reduce transcription-promoting mediator complexes, as the increased oxidative stress in PCa cells [185,186] may result in CDK8 activation, which causes CDK8 to release from the mediator complex [187,188]. The increase of CDK8 expression in CRPC may also have a role in the increase in WNT signalling [182], a pathway which is almost always inactive in normal prostate cells but is often dysregulated in CRPC [190] and is linked to ADT [191] and chemotherapy resistance [192]. Furthermore, WNT signalling downregulates cadherin, a key molecule in cell-cell adhesion, and is implicated in metastasis and the epithelial-mesenchymal transition [193]. Many studies have looked at genetic alterations of the WNT pathway; however, aberrant WNT signalling can also occur through other mechanisms [193]. Β-catenin is a key effector molecule of canonical WNT signalling and is inhibited and degraded in multiple ways, including by E2F1, which CDK8 phosphorylates and inhibits [194]. Thus, the increased level of β-catenin in the cytoplasm of prostate cancer cells [193] may be partially mediated by increased CDK8 expression. Β-catenin can interact with AR in its active conformation to increase androgen-dependent AR transactional activity [195] and β-catenin expression [196] to create a positive feedback loop. WNT/β-catenin signalling increases AR expression [197], which may perpetuate the AR-β catenin-positive feedback loop in CRPC. Therefore, CDK8 may be a therapeutic target for CRPC, or to inhibit alongside ADT. CDK8 and its analogue CDK19 regulate distinct gene profiles, and both may need to be inhibited for the optimum therapeutic benefit [182]. Indeed, several CDK8/CDK19 inhibitors are in preclinical development, and one—BCD-115—has completed Phase I clinical trials (Trial identifier: NCT03065010), although the results have not yet been posted [198]. Recently, Ma et al. reviewed the binding profiles of 24 CDK8 inhibitors and highlighted four methods of identifying new inhibitors, which may be more selective for CDK8 [199].

### 3.5. Splicing Factors

GO analysis of the kinases we previously found to be downregulated [14] showed enrichment for nuclear speck organisation, due to Dual Specificity Tyrosine Phosphorylation Regulated Kinase 3 (DYRK3) and SRSF Protein Kinase 2 (SRPK2) repression. The function of nuclear specks is debated; however, they consist of high concentrations of Serine and arginine-rich (SR) proteins and other splicing factors [200]. DYRK3 and SRPK2 both cause the dissolution of nuclear speckles [201,202].

Indeed, SRPK2 overexpression in PCa is associated with high Gleason grade and metastasis in AR-dependent and AR-independent cell lines, indicating SRPK2 may have a role in the progression of PCa and initiation of metastasis [203]. In androgen ablation, PI3K/Akt/mTOR1 signalling increases [75], which results in SPRK2 phosphorylation and translocation to the nucleus [202]. In the nucleus SRPK2 promotes disassembly of nuclear speckles by phosphorylating certain splicing factors, including ASF/SF2, resulting in translocation out of the nuclear speckles, where they associate with nascent mRNA [202]. Lui et al. found that ASF/SF2 was recruited to 3′ splice sites on AR pre-mRNA for AR-V7 during ADT with enzalutamide [204]. Therefore, the increased expression of SRPK2 due to androgen ablation could be a mechanism by which AR-V7 transcripts increase and the subsequent issues with ADT resistance. SRPK2 inhibition could potentially re-sensitise CRPC to ADT through the reduction in AR-V7, however, the roles of SRPK2 and DYRK3 in PCa and splicing of the AR requires further investigation.

## 4. Clinical Implications and Future Perspectives

Several kinase inhibitors have been tested for in a large range of human malignancies, and a search on the clinicaltrial.gov database (accessed on 30 January 2022) with the terms “prostate cancer” and “kinase inhibitor” returned 106 clinical trials that were either recruiting, active, or completed. Looking forward, Table 1 lists 20 active or recruiting clinical trials, and which of the functional pathways or groups discussed in the review are targeted. The SMMART and MATCH clinical trials were excluded as kinase inhibitors were given were dependent on the genotype of the patient.

## 5. Conclusions

Kinases are heavily involved in the regulation and biological effects of AR activation and are interrelated in a complex robust network, where AR-responsive genes also affect cascades upstream of the AR and the AR directly. Kinases upstream of the AR, such as ACK1 and Src, present as therapeutic targets to reduce androgen-independent activation of the AR due to extracellular signalling by EGF and IL-6. Androgen-responsive kinases, which are upregulated, present as targets to reduce the proliferation of AR-dependent prostate cancer, and downregulated kinases imply changes that can occur during ADT and, thus, may be therapeutic targets for reducing the progression of androgen independence of prostate cancer. Some of the androgen-responsive kinases are under-researched and could present as therapeutic targets, such as COASY and SRPK2.

## Figures and Tables

**Figure 1 cells-11-00952-f001:**
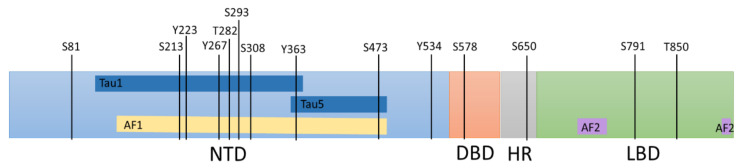
A schematic summarising the phosphorylated sites within the androgen receptor which moderate its activity. The androgen receptor contains several serine (S), threonine (T), and tyrosine (Y) residues that can be phosphorylated to moderate localisation, stability, and transactivation activity. Amino terminus transactivation domain (NTD), the DNA binding domain (DBD), the Hinge region (HR), and the ligand-binding domain (LBD). The activation function 1 (AF1) region (yellow line) consist of the majority of Tau1 and Tau5 (dark blue lines). The activation function 2 (AF2) region (purple lines) in the LBD is primarily located in the 12th helix; however, parts of the 3rd and 4th helixes also make up the AF2 surface [33].

**Figure 2 cells-11-00952-f002:**
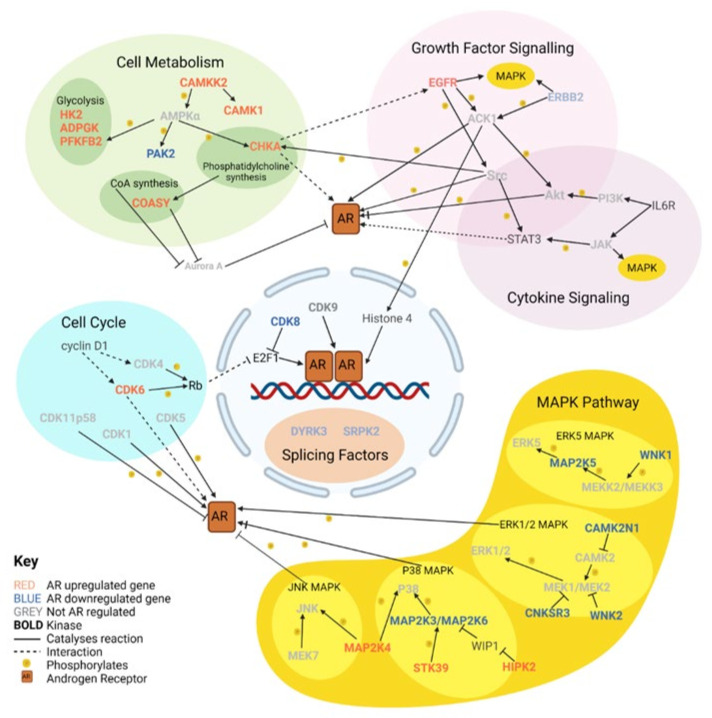
Key cellular pathways involving upstream and downstream kinases of the androgen receptor. Different functional groups are contained within different coloured ovoids. Within Cell Metabolism and the MAPK Pathway are subgroups represented by further coloured ovals. There are 24 AR-regulated kinases that are discussed in the main text, with other key kinases and molecules these kinases interact with included.

**Table 1 cells-11-00952-t001:** Overview of active or recruiting clinical trials of kinase inhibitors for prostate cancer.

NCT Number	Small Molecules	Target Kinases	Target Pathway	Phase	Status
NCT04159896	CEP-11981 (with Nivolumab)	VEGFR, TEK	Growth Factor Signalling	Phase 2	Recruiting
NCT03456804	CEP-11981	VEGFR, TEK	Growth Factor Signalling	Phase 2	Active
NCT05000294	Atezolizumab, Tivozanib	VEGFR(Tivozanib)	Growth Factor Signalling	Phase 1, Phase 2	Recruiting
NCT02893917	Cediranib, Olaparib	VEGFR (Cediranib)	Growth Factor Signalling	Phase 2	Active
NCT04848337	Pembrolizumab, Lenvatinib	VEGFR(Lenvatinib), FGFR (Lenvatinib)	Growth Factor Signalling	Phase 2	Recruiting
NCT00329043	Sunitinib Malate, LHRH Agonist	VEGFR, PDGFR, KIT, CSF1R, RET	Growth Factor Siganlling	Phase 2	Active
NCT01409200	Axitinib	VEGFR, ABL	Growth Factor Signalling	Phase 2	Active
NCT04140526	ONC-392, Pembrolizumab, Osimertinib	EGFR(Osimertinib)	Growth Factor Signalling	Phase 1	Recruiting
NCT02484404	Durvalumab, Cediranib, Olaparib	VEGFR(Cediranib)	Growth Factor Signalling	Phase 1, Phase 2	Recruiting
NCT04925648	Darolutamide, Dasatinib	Src (Dasatinib), ACK1 (Dasatanib)	Growth Factor Signalling, Cytokine Signalling.	Phase 2	Recruiting
NCT04869488	Fluzoparib, Apatinib	VEGFR(Apatinib), KIT (Apatinib), Src(Apatinib)	Growth factor Signalling, Cytokine Signalling	Phase 2	Recruiting
NCT01254864	Abiraterone Acetate, Prednisone, Sunitinib, Dasatinib	VEGFR (Sunitinib), PDGFR (Sunitinib), KIT (Sunitinib), Src (Dasatinib), ACK1 (Dasatinib)	Growth Factor Signalling, Cytokine Signalling	Phase 2	Active
NCT01990196	Degarelix, Enzalutamide, Trametinib, Dasatinib	Src (Dasatinib), ACK1 (Dasatinib), MEK1 (Trametinib), MEK2 (Tramentinib)	Growth Factor Signalling, Cytokine Signalling, MAPK Pathway	Phase 2	Active
NCT02881242	Trametinib	MEK1, MEK2	MAPK Signalling	Phase 2	Active
NCT03414034	Onvansertib, Abiraterone, Prednisone	PLK1(Onvansertib)	Cell Cycle	Phase 2	Recruiting
NCT04267939	BAY1895344, Niraparib	ATR kinase (BAY1895344)	Cell cycle	Phase 1	Recruiting
NCT04071236	Avelumab, Peposertib	DNA-PK	Other	Phase 1, Phase 2	Recruiting
NCT04606446	PF-07248144, Fulvestrant, Letrozole with Palbociclib	CDK4/6 (Palbociclib)	Cell Cycle	Phase 1	Recruiting

## Data Availability

Not applicable.

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
