# Peer review of "Unravelling the Role of Kinases That Underpin Androgen Signalling in Prostate Cancer"

_cells, 2022, doi:10.3390/cells11060952_

Round 1
Reviewer 1 Report
The manuscript here presented is very intriguing. It is focused on the role of androgen receptor (AR) in Prostate cancer (PC) but considers different aspects. It is well written and pictures are well performed.
I have only some suggestions for expanding the news with recent papers:
- The authors consider AR as a master transcription factor, without hinting that it can be also involved in non-genomic actions. It happens in the Prostate cancer microenvironment for example and in particular, in carcinoma-associated fibroblasts (CAFs) derived from patients affected by PC at different Gleason's scores. The receptor is localized in the cytoplasmic compartment and it has no transcriptional acivity, irrispective of androgen stimulation also if it is wild type. (Cell Death and Disease, 2021)
- In the paragraph" Growth Factor Signalling": the authors do not discuss the cross-talk between AR and TrkA (NGF receptor) which occurs in LNCaP cells. There is a reciprocal crosstalk which affects also the proliferation and migration of PC cells. This is a novelty but also draws to the concept of plasticity of PC. Again, After NGF stimulation, Trk A phosphorylation mediates also ERK and Akt kinaese activation in CRPC (expressing or not expressing AR) through rapid and non genomic actions. (cell death and discovery, 2018 and cancers 2019)
Author Response
please see attached the revised cover letter containing a point-by-point response to comments made by the reviewers.

Reviewer 2 Report
The authors conducted a comprehensive review of the kinase upstream and downstream the androgen receptor. This review covers an interesting topic in prostate cancer research, suggesting potential novel therapeutic targets and the development of kinase inhibitors for prostate cancer.
The manuscript is generally well-written, mainly informative and satisfactory.
Current review may pose an interest to the readers. However, the authors should address some questions that appeared during the manuscript review. Please see the comments below.
- In the introduction I suggest to give some clinical information about prostate cancer treatment. During the last years several improvements in treatment strategy have been made. The androgen receptor has been historically considered the most important target to control prostate cancer growth and most drugs are directed against AR pathway. Androgen deprivation therapy represent the standard of care in advanced prostate cancer. In the treatment of metastatic HSPC the addition of docetaxel, abiraterone, enzalutamide or apalutamide to ADT is supported by recent data. However, the hormone sensitive phase in metastatic patients is only transient and most patients eventually develop metastatic castration resistant prostate cancer after a median of 18-36 months. Metastatic castration resistant prostate cancer can be treated with chemotherapy (docetaxel and cabazitaxel), AR pathways inhibitors (abiraterone and enzalutamide), the alpha emitters Radium 223 or PARP-inhibitors.
- Before the description of the role of kinase that underpin androgen receptor signaling in prostate cancer, it may be useful to briefly describe the main mechanisms of resistance to hormonal therapy, both AR dependent and AR-independent, in order to highlight the need of new therapeutic agents able to overcome resistance.
- In paragraph 2.5 (Cell cycle-dependent modulation of AR activity) I suggest to mention the role of Rb1 in prostate cancer, and highlight that Rb loss is reported in about 20% of prostate cancer and its deletions are significantly more common in CRPC, especially metastatic castration resistant prostate cancer and neuroendocrine prostate cancer, than in primary site. Rb inactivation might compromise the efficacy of antiandrogenic therapy, such as next generation hormonal therapies. Moreover, in vitro data suggest that Rb1 loss may enhance taxane activity (which represent a cornerstone in the treatment of metastatic castration resistant prostate cancer and high volume metastatic hormone sensitive prostate cancer).
- please check the references (i.e: ref 52b, line 159)
- I suggest to add a brief paragraph to summarize the clinical implications and future perspectives and also a table of the ongoing trials testing drugs targeting each of the pathways described in the text.
Author Response

(The authors gave the same response as above.)

Round 2
Reviewer 1 Report
Thr manuscript is improved